# Chemical Characterization of Clove, Basil and Peppermint Essential Oils; Evaluating Their Toxicity on the Development Stages of Two-Spotted Spider Mites Grown on Cucumber Leaves

**DOI:** 10.3390/life12111751

**Published:** 2022-10-31

**Authors:** Salonaz E. Awad, Karima Bel Hadj Salah, Muthana M. Jghef, Abeer Mousa Alkhaibari, Ashjan A. Shami, Rana Abdullah Alghamdi, Ramadan M. El-Ashry, Abdelhadi A. I. Ali, Lamiaa M. M. El-Maghraby, Ahmed E. Awad

**Affiliations:** 1Department of Plant Protection, Faculty of Agriculture, Zagazig University, Zagazig 44511, Egypt; 2Biological Sciences Department, College of Science & Arts, King Abdulaziz University, Rabigh 21911, Saudi Arabia; 3Laboratory of Transmissible Diseases and Biologically Active Substances, Faculty of Pharmacy, University of Monastir, Monastir 5019, Tunisia; 4Department of Radiology, College of Medical Technology, Al-Kitab University, Kirkuk 36001, Iraq; 5Department of Biology, Faculty of Science, University of Tabuk, Tabuk 71491, Saudi Arabia; 6Department of Clinical Laboratory Sciences, College of Applied Medical Sciences, Taif University, Taif 21944, Saudi Arabia; 7Department of Chemistry, College of Sciences & Arts, King Abdulaziz University, Rabigh 21911, Saudi Arabia; 8Department of Agricultural Biochemistry, Faculty of Agriculture, Zagazig University, Zagazig 44511, Egypt

**Keywords:** essential oils, cucumber cultivars, *Tetranychus urticae*, biological aspects

## Abstract

**Simple Summary:**

Essential oils show considerable acaricidal activity against pests. This study investigated the effects of three essential oils on *T. urticae*, one of the most serious pests in the world. The quality of the host plant and growth conditions affect the reproduction of *Tetranychus urticae* Koch, reducing their population levels very quickly. Research is ongoing to find eco-friendly insecticides or natural bioactive compounds against spider mites under greenhouse and field conditions. Clove EO was found to be the most toxic, while basil and peppermint EOs were the least effective, and immature stages were more sensitive to EOs than mature stages. It can be concluded that cucumber cultivars are effective in regard to the biological aspects and reproduction of two spot spider mites (TSSM), and the tested oils are good alternatives to control *T. urticae* in the protected cultivation of cucumbers.

**Abstract:**

The two spotted spider mite (TSSM), *Tetranychus urticae* Koch, is a cosmopolitan mite. It rapidly reproduces and can develop resistance to chemical pesticides. This study aims to evaluate the toxicity and acaricidal activity of three essential oils from basil, clove, and peppermint against *T. urticae* reproduction, which is grown on three cucumber cultivars, Chief (SC 4145), Raian (CB898), and Toshka (SC 349), under laboratory conditions at 27 + 3 °C and 70 + 5% RH. GC-MS characterized the volatile oils of basil, clove, and peppermint. Methyl cinnamate, eugenol, and menthol were the main essential oils in basil, clove, and peppermint, respectively. The results indicated significant differences in the duration of development between *T. urticae* feeding on the three cucumber cultivars (*p* ≤ 0.05), including eggs, protonymph, and deutonymph time. The Toshka (SC 349) cultivar recorded the lowest developmental time. The longevity period exhibited the same trend with non-significant differences between Raian (CB898) and Toshka (SC 349). Moreover, the lethal concentration (LC50) and LC90 values in tested essential oils (EOs) showed that clove EOs were the most toxic. In contrast, basil and peppermint EOs were the least effective, and immature stages were more sensitive to EOs than adult stages. The infected Toshka (SC 349) discs treated with essential oils and abamectin under in vitro conditions indicated that clove oil is comparable to abamectin regarding its effect on the egg numbers (18.7 and 17.6 egg), immature development time, longevity, life span, and life cycle (20.6 and 20.8 days) of *T. urticae*. We conclude that the resistant cultivation of cucumber plants can be recommended in integrated pest management programs. The most effective of the tested oils, clove EOs, should be used as alternatives to pesticides to control *T. urticae* in the protected cultivation of cucumbers.

## 1. Introduction

Natural products such as essential oils (EOs) have been considered good alternatives to synthetic pesticides because of their lethal or sublethal effects, such as miticidal, repellent, and oviposition deterrent activities, for numerous pests and low mammalian toxicity [1,2,3,4,5]. Plant essential oils are typically composed of complex mixtures of mono-and sesquiterpenoids, such as 1,8-cineole, eugenol, and menthol [6,7]. Owing to the structural diversity of chemical constituents, major constituents in the same oil or the mixture of essential oils interact synergistically or antagonistically, and sometimes simply additively [8,9,10,11]. Understanding and identifying these complex relationships is critical for using botanical sources in pest control [12,13,14].

The two-spotted spider mite (TSSM) (*Tetranychus urticae* Koch (Acari: Tetranychidae)) wreaked havoc on fruit, vegetable, and ornamental plants all over the world [15]. As a result of its high reproduction rate, short growth period, low humidity, and high temperatures, its population quickly increases. Moreover, with the lack of farmers’ knowledge of biology and proper control programs, this mite has recently caused severe damage to vegetables and the yields of fruit crops [16]. Furthermore, reevaluating all synthetic pesticide tolerances, particularly acaricides, which impede the global control of *Tetranychus* species [17], in addition to the growing public concern about environmental pollution and health hazards caused by synthetic pesticides creates a great need for new classes of pest control agents with higher activity against the target pests and lower impact on humans and environmental quality [18]. Therefore, there is a need to develop an effective eco-friendly pesticide or bioactive natural compounds with low natural enemies and mammalian toxicity to control vegetable insects and spider mites under greenhouse and field conditions. Nowadays, recent control technology and proper tactics have been used to achieve the sustainable management of environmental problems either in vitro or under field conditions [19,20].

*Tetranychus urticae’s* rapid developmental rate, short generation time, and high net reproductive rate allow them to achieve damaging population levels very quickly when growth conditions are good, resulting in an equally rapid decline of host plant quality [21,22,23]. Reproduction in spider mites is very sensitive to a wide variety of intrinsic and extrinsic conditions used in their biological control [24]. Various plant extracts of numerous medicinal plants were tested against *T. urticae* to assess their effects on some biological aspects and the productivity of spider mites [25,26,27].

Many acaricides, including abamectin and fenopyroximate, have been widely used for *T. urticae* control in open fields as well as in glasshouses [28,29,30]. The high reproductive potential and short life cycle of *T. urticae* combined with the frequent application of acaricides lead to the development of resistance [31].

Therefore, the need for new effective, biodegradable, and safer biopesticides has increased dramatically. The global biopesticide market has grown from $1.6 billion in 2009 to $3.3 billion in 2014 at a 15.6% compound annual growth rate (CAGR) [32]. The use of natural products to control agricultural pests has intensified in recent years [33]. The search is continuous for substances with acaricidal properties as alternatives to conventional pesticides for use in the integrated management of *T. urticae* [34]. Moreover, there are a few studies about the acaricidal activity of the chemical constituents of essential oils and their role in the biological activity of such oils. So, different chemical components can be mixed to make synthetic blends by choosing the best combination of compounds for use in the integrated management of the two-spotted spider mite and looking into how much each compound contributes to the synthetic oil [35]. The essential oils demonstrate nearly no side effects on plants, humans, and the environment. 

So, the current study aimed to use three cucumber cultivars, namely Chief (SC 4145), Raian (CB898), and Toshka (SC 349), as a host for *Tetranychus urticae* Koch (Acari; Tetranychidae) by considering their development and longevity and evaluating the toxicity effect of three essential oils, composed of basil leaves (*Ocimum basilicum*), leaves of peppermint (*Mentha piperita*), and clove buds (*Syzygium aromaticum*), on the biological aspects and fecundity of this pest compared to abamectin in vitro, to choose the most effective among the tested oils for controlling this pest and the host suitability so that it can be used safely in the context of integrated pest management. 

## 2. Materials and Methods

### 2.1. Materials

The TSSM individuals were reared on pepper (*Capsicum annuum* L.) plants, cultivar 1515, grown in plastic pots (15 cm in diameter and 21 cm in height) in a growth chamber (27 + 3 °C, 75% RH, and a photoperiod of 16.5 L: 7.5 D h.) for several generations (at least three months). All experiments were performed under the conditions mentioned earlier in growth chambers.

Experiments on the fecundity and development time of *T. urticae* were conducted on three cucumber cultivars: Chief (SC 4145), Raian (CB898), and Toshka (SC 349). The seeds of three cucumber cultivars were sown in plastic pots (15 cm in diameter × 21 cm in height) filled with sterilized media (sand: clay, 1:1), and normal agriculture practices, including irrigation and fertilization, were applied. After 17 days, cucumber leaves were detached and used for leaf disc preparation. All plants were irrigated simultaneously during the experiments, and no pesticides were used.

### 2.2. Experimental Design

The leaf disc method was used according to Naher et al. [36]. Each leaf disc had a 5 cm^2^ area cut from the center of the leaves. Each leaf disc was placed on a plastic Petri dish (9 cm in diameter × 1.5 cm high with a hole in its center). After that, one fully expanded young leaf (third leaf below the apical meristem of 3 weeks old plants) was randomly collected and used for the leaf disc preparation. The leaves of various cucumber cultivars were selected from all replications, cut into a leaf disc (2.5 × 2.5 cm), and then placed on water-saturated cotton in the Petri dish with the underside facing upward. During the tests, cucumber seeds were planted every week to reduce the effect of plant age on mite growth and reproduction and to provide the mites with new leaf discs.

### 2.3. Biological Tests

#### 2.3.1. Construction of the Fertility Life Table and Determination of Developmental Time

The reproductive and growth parameters of *T. urticae* were determined on three cucumber cultivars in laboratory conditions at 27 ± 3 °C, 70 ± 5% RH and a photoperiod of 16.5 L: 7.5 D h. To construct a fertility life table for each treatment and determine the developmental time of the immature stages of *T. urticae* on each cultivar, a cohort of 100 eggs was individually followed. Each rearing unit consisted of an egg (less than 2 h old) placed upside down on water-saturated cotton wool in a 6-cm diameter Petri dish, taken from a leaf detached from the terminal shoots (with three leaves) of each cultivar. Observations were made twice daily, and the duration of the development of the egg, larva, protonymph, deutonymph, quiescent stages, adult longevity, and fecundity of the mite were recorded for each cultivar. After the emergence of a female, one male was introduced to each test unit. Eggs laid per female were recorded daily. Observations continued until the deaths of all individual members of the cohort. Three random samples (at least 60 eggs in each sample) were taken from the eggs laid at the beginning, middle, and near the end of the female’s oviposition period to determine the sex ratio. The collected eggs were allowed to develop into adulthood, and their gender was determined. The rearing units were kept in a growth chamber at a constant 27 + 3 °C, 70 + 5% RH, and 16.5 L: 7.5 D h photoperiod.

#### 2.3.2. Essential Oil Isolation and Identification

##### Isolation by Steam Distillation

In this study, three herbal and medicinal plant species were selected according to their ethnomedicinal importance and literature survey. The leaves of basil *(Ocimum basilicum*) and peppermint (*Mentha piperita*) and clove (*Syzygium aromaticum*) buds were dried in an oven at 45 °C for 5 days, ground then powdered. 100 g of each plant powder were subjected to essential oil extraction. Oils were extracted through steam-distillation using a Clevenger-type apparatus [37]. After a distillation time of 3 h, 100 g of each plant of the dried material was yielded, as demonstrated in Table 1. The distillation was repeated to obtain the required oil quantity for research purposes. 

Three EOs of basil, peppermint and clove buds were tested against the two-spotted spider mite, *Tetranychus urticae* Koch (Acari: Tetranychidae) in vitro and compared to Abamectin.

##### Identification of EOs by GC-MS 

For determination of the chemical composition of basil leaves, leaves of peppermint and clove bud EO, gas chromatography-mass spectrometry (GC-MS) analysis was conducted using GC-2010 Shimadzu capillary gas chromatography directly coupled to the mass spectrometer system (GC-MS–model QP 2010; (Shimadzu, Kyoto, Japan) DB-c18 column under the following conditions: The injector temperature is 250 °C. Oven temperature program: 30 °C for 2.0 min, then ramp to 250 °C at a rate of 2.0 °C per minute for 5.0 min. The MS source temperature was 200 °C, electron energy was 70 eV, the carrier gas was helium at a flow rate of 1.4 mL/min, and 1 µL of each diluted sample in n-hexane (1:1, *v*/*v*) was injected. EI spectra were scanned from 43.00 to 600 *m*/*z* to identify peaks through NIST mass data search libraries and the highest REV and similarity indicators hits. The sample components were identified by comparing their relative indices and mass spectra with the computer controlling the GC-MS system [38].

### 2.4. Toxicity Test

Toxicity tests in vitro were performed in plastic boxes, each containing a leaf of cucumber (*Cucumis sativus* L.) Chief (SC 4145) cultivars were placed on water-soaked cotton. Sequential concentrations of essential oils (0.25, 0.50, and 0.75 standard solution (SS) were used against immature stages (egg, larva, protonymph and deutonymph) of *T. urticae*. After spraying the essential oils with a manual spray (spray bottle) of essential oils per leaf, all leaves were dried for 5 min. Next, immature stages of TSSM were introduced into each Petri dish (5 individuals) in five plastic dishes and five adult stages (female or male) in 10 other boxes (5 dishes).

The numbers of dead mites were calculated at 24, 48, and 72 h. After post-treatment, mites were considered dead when probing with the tip of a fine-haired brush; no appendages moved, looked black, or eggs were non-hatching. After three days, the following equation was used to calculate the mortality percentage of mite larvae
$$\mathrm{Mortality}\;\left(\%\right)=\frac{\;\mathrm{Number}\;\mathrm{of}\;\mathrm{dead}\;\mathrm{mite}\;\mathrm{stage}}{\mathrm{Total}\;\mathrm{mite}\;\mathrm{stage}}\times 100$$

For each treatment, five replications were realized. The Abamectin at recommended rate (RC) 15 mL/100Lwas used as a reference to be compared with the treatments using essential oils [39].

#### LC_50_ and LC_90_ for Adult Males and Females of *T. urticae*

Lethal concentration 50 (LC_50_) is the concentration of active compound required to kill 50% of TSSM, and LC_90_, is the is the concentration of active compound to kill 90% of TSSM. The leaves of cucumber plants were punched for preparing leaf discs (3.0 cm in diameter). Leaves discs were placed on wet cotton pads in Petri dishes (9 cm diameter). A-concentrations (i.e., 2.5, 5.0, 10.0, 20.0, 40.0, 80.0, and 100.0 mL/L) of essential oils were replicated five times.

One mL of tested concentrations was sprayed on leaf discs with a hand sprayer for even and complete coverage. Control treatments were made using tap water and the spreader only. After drying leaf discs at room temperature, ten adult males or females of *T. urticae* were transferred to the lower surface of each treated leaf disc. *T. urticae* individuals were exposed to treated leaf discs for 24, 48, and 72 h. Mortality percentages were calculated after 24 h, 48 h, and 72 h. Mite mortality was counted according to the mentioned equation and dead signs.

### 2.5. Statistical Analysis

Regression toxicity lines were established for the pesticides and the slope. LC_25_ and LC_50_ values were determined through probit analysis [40]. Data statistical analysis was performed using SPSS version 20.0. Data of the biological studies were analyzed using the (ANOVA) test and the means were analyzed using the LSD test at (*p* ≤ 0.05) [41].

## 3. Results

### 3.1. Essential Oil Composition

The results of the oil yield essential oils investigated are given in Table 1. The oil yield in terms of the percentage of basil, clove buds, and peppermint oils was 2%, 1.5%, and 0.8%, respectively. According to GC-MS, 24 constituents were identified in the basil oil, accounting for 98.36 % of the total compositions. The main volatile compounds (VOCs) of peppermint were methyl cinnamate (41.39%), linalool (32.5%), and 1,8-cineole (6.18%) (Table 2). Approximately 97.8 % of VOCs were identified in the clove bud oil. The major VOCs of clove detected were Eugenol (50.2%), Caryophyllene (19.3%), and Isoeugenol (16.7 %) (Table 3). The VOCs of the peppermint oil represent about 98.01% of the total compositions identified. The main volatile constituents of peppermint detected were menthol (42.13%), menthone (22.82%), mentho-furan (11.79%), and 1,8-cineole (4.1%) (Table 4).

### 3.2. Influence of Host Plants on T. urticae

#### 3.2.1. Effect on Immature Development Time

The present investigations focused on the effects of host plant cultivars on biological aspects of *T. urticae* under laboratory conditions (Table 5). Current results indicated the biological tests of *T. urticae*, including immature developmental stages, adult longevity, generation time, and life span with leaves of three cucumber plant cultivars, Chief (SC 4145), Raian (CB898) and Toshka (SC 349), in Petri dishes at 27 ± 3 °C and 70 ± 5% R.H. A successful development from *T. urticae* eggs to adult emergence was conducted to compare immature development time, life span, and sex ratio. There were significant differences in the duration of development between *T. urticae* feeding on the three different cucumber cultivars (*p* ≤ 5.01). *T. urticae eggs* were hatched after 3.34, 3.15, and 3.10 days, respectively, and clearly, Toshka (SC 349) cultivar, with accelerated hatching, was selected for its vulnerability to mites. Moreover, larvae stage time ranged from 3.16 days on leave discs of Toshka (SC 349) to 3.23 days in those reared on Raian (CB898) and 3.41 days on discs of Chief (SC 4145), respectively.

The cucumber plant cultivar Chief (SC 4145) had the longest protonymoh and deutonymph times, with 3.25 and 2.94 days, respectively, followed by 3.21 and 2.71 days with the Raian (CB898) cultivar and 3.06 and 2.71 days with the Toshka (SC 349) cultivar. There was no significant difference in immature development time from eggs to deutonymph between the Raian (CB898) cultivar (12.30 days) and the Toshka (SC 349) cultivar (12.24 days). In contrast, there was a significant difference in the immature development time reared on Toshka (SC 349). 

#### 3.2.2. Effect on Generation Time, Life Span and Sex Ratio 

The results indicated that the pest generation time from the successful rearing of cucumber plant cultivars was obtained from the leaf discs of Toshka (SC 349) cultivar (25.12 days) followed by Raian (CB898) cultivar (26.25 days) with an insignificant difference (*p* ≤ 0.05), while generation time was prolonged to reach 27.97 days with Chief (SC 4145) cultivar. The same trend was observed with the calculated life span of *T. urticae* reared on three tested cucumber cultivars. For example, the life span of *T. urticae* reared on Raian (CB898) and Toshka (SC 349) was 13.85 and 13.59 days compared with 14.90 days with TSSM, *T. urticae* reared on Chief (SC 4145), respectively. In terms of sex ratio, TSSM reared on Toshka (SC 349) had a higher female ratio than Chief (SC 4145) and Raian (CB898), with sex ratios of 1:5.71, 1:4.12, and 1:4.37, respectively.

Generally, from the obtained results, it can be concluded that *T. urticae* varied when reared on Chief (SC 4145), Raian (CB898), and Toshka (SC 349) cucumber cultivars in terms of immature development time as well as generation, life span, and sex ratio under laboratory conditions.

#### 3.2.3. Effect on Longevity of *T. urticae*

Statistical analysis showed that there were highly significant differences between three cucumber cultivars in periods of pre-oviposition, oviposition, and post-oviposition periods and longevity of *T. urticae.* The pre-oviposition period ranged from 1.91 days with cultivar Chief (SC 4145) to 1.55 and 1.35 days with Raian (CB898) and Toshka (SC 349) cultivars, respectively. At the same time, the effect of cucumber cultivars on the oviposition period was assessed as 10.63, 9.96, and 9.21 days with Chief (SC 4145), Raian (CB898), and Toshka (SC 349), respectively. Moreover, the post-oviposition period in *T. urticae* reared on the three mentioned cultivars was 2.44, 2.39, and 2.27 days, respectively. The longevity period followed a similar pattern, with 14.98 days, 13.95 days, and 12.88 days in Chief (SC 4145), Raian (CB898), and Toshka (SC 349) cultivars, respectively, with non-significant differences in Raian (CB898) and Toshka (SC 349) cultivars (Table 6). 

### 3.3. Influence of Different Essential Oils Concentrations on Two—Spotted Spider Mite

#### 3.3.1. Effect on Adults of Two—Spotted Spider Mite *T. urticae*

The effect of clove, basil, and peppermint EOs on the mortality of adult two-spotted spider mite *T. urticae* was significantly (*p* ≤ 0.05) diverse according to varied concentrations of essential oils when compared with control (Table 7). Moreover, LC_50_ and LC_90_ were assessed with three tested oils post-treatment. The obtained results in Table 7 show that clove oil was the most effective when compared with basil and peppermint oil in controlling *T. urticae* adults. Among the tested concentrations, the concentration of 200 mL L^−1^ of clove oil showed a significantly superior effect (100% mortality) against the adults of *T. urticae* compared to the same concentration with basil and peppermint after 24 h. After 48 h and 72 h of treatment, the concentration of 150 resulted in 84.20% and 90.00% mortality in Petri dishes treated with clove oil compared with 62.60% and 66.80% and 64.2% and 68.60% in Petri dishes treated with basil and peppermint, respectively. On the other hand, clove oil concentration (50 µLL^−1^) showed more than 50% mortality after 24 h of treatment compared with 38.20% and 48.80% after 72 h of treatment with basil and peppermint, respectively. LC_50_ of clove, basil, and peppermint essential oil was 46.80, 138.40 and 68.80 µLL^−1^, respectively, while LC_90_ values were 150.00, 442.20 and 325.00 µLL^−1^ with the three mentioned oils respectively. This indicates that clove oil is more effective against the adults, because a small concentration is enough to kill 50% of the *T. urticae* mites followed by peppermint while basil was the least effective. 

#### 3.3.2. Effect on Immature Stages of Two—Spotted Spider Mite *T. urticae*

The same trend was observed when we tested the effect of clove, basil oil, and peppermint oils on the mortality of immature stages of two-spotted spider mites, *T. urticae.* Clove oil was significantly (*p* ≤ 0.05) the most effective one compared with control basil and peppermint oils (Table 8). Moreover, LC_50_ and LC_90_ for killing immature stages of *T. urticae* showed that clove oil’s toxicity surpassed that of the two tested basil and peppermint oils under in vitro conditions. From the current results in Table 8, clove oil (150 and 200 µLL^−1^), achieved mortality percentages of 74.80% and 80.20% after 24 h compared with that of basil oil (48.20% and 56.40%) and peppermint oil (56.80% and 70.20%). After 48 h and 72 h of clove oil (150 µLL^−1^) treatment, the % mortality increased to 80.20% and 88.00% compared with (60.20% and 66.00%) for basil and (62.60% and 66.60%) for peppermint oil at the same concentration. On the other hand, immature stage mortality at a concentration of 50 was 60.20% after 72 h of clove oil treatment, compared with 36.40% and 48.00% for basil and peppermint treatments, respectively. LC_50_ values were 58.80, 158.80, and 125.40 for clove, basil, and peppermint oils, while LC_90_ was 186.72, 492.78, and 396.20 with the three mentioned oils, respectively. The immature stages of *T. urticae* were the most sensitive to essential oils at the same tested concentrations. Based on the results in Table 7 and Table 8, the LC_50_ of the adults, i.e., 46.80 µLL^−1^, is smaller than the immature stages, i.e., 58.80 µLL^−1^, which indicates that the oil is more effective against the adults, because a small concentration is enough to kill 50% of the mites. 

### 3.4. Developmental Periods (in Days) of Tetranychus urticae Females Reared on Leaves of Cucumber Plant Toshka (SC 349) Cultivar Treated with Essential Oils

#### 3.4.1. Effect on Egg Deposition

The effects of EO stock solutions (SS) of plant extracts and the recommended rate of abamectin on eggs deposited by the adult female mites of *T. urticae* were studied. Moreover, their effects on the immature development time, life span, and life cycle of *T. urticae are unclear.*

As shown in Table 9, five adult female mites of *T. urticae* feeding on leaf discs of the cucumber plant Toshka (SC 349) cultivar treated with different materials and abamectin were allowed to oviposit on leaf discs for five days. Eggs were counted daily for five days, and different cucumber leaf discs were replicated five times (Table 9). When comparing between untreated and untreated Toshka (SC 349) with accumulative eggs deposited by the adult females of mite *T. urticae*, results showed that abamectin was the most effective compound on egg deposition with a general mean of 17.80 eggs, followed by clove oil (18.76), *M. azedarach* (19.64), and peppermint oil (19.96). Basil oil (22.88) was the least effective of the tested compounds.

#### 3.4.2. Effect on Life Cycle and Life Span

As shown in Table 10, successful rearing of *T. urticae* on cucumber plant cultivar Toshka (SC 349) treated with tested materials revealed that the immature development time of *T. urticae* feeding on abamectin was 14.99 days, followed by clove oil (14.06 days) and basil oil (13.69 days). Whereas the immature development time that resulted from treatment of peppermint oil was 13.22 days. The same trend was observed with the calculated life span of *T. urticae* reared on treated cucumber cultivar discs with the mentioned materials. For example, the life span of *T. urticae* reared on discs treated with abamectin was 31.68 days, followed by clove oil (30.88 days), compared with 29.00 days with TSSM, *T. urticae* reared on peppermint oil discs. From the current results, abamectin was the most effective in the life cycle and life span periods (20.82 and 31.68 days), followed by clove oil (20.66 and 30.88 days), with insignificant differences (*p* ≤ 0.05).

## 4. Discussion

Various factors influence the biological aspects of TSSM, such as plant cultivars or food quality. In this regard, Awad et al. [42], Awad [43] tested the effect of solanaceous vegetable crops on the development and reproduction of *T. urticae,* with results that logically displayed variation in the generation time and life span of *T. urticae* reared on cucumber cultivars at 27 ± 3 °C, 75 ± 5% RH [42,44]. The host plant’s quality directly affects the development and survival of TSSM, *T. urticae* Koch. On the other hand, current results revealed that the quality of the host plant affects TSST development and reproduction and has a critical effect on the sex ratio of *T. urticae* reared on cucumber cultivars. The sex ratio of TSST reared on Toshka (SC 349) was 1:5.71, followed by Raian (CB898). Likewise, the pre-oviposition, oviposition, and post-oviposition periods were significantly shortened in TSST reared on Toshka (SC 349) compared to other cultivars. The biology and fertility life tables of *T. urticae* differed significantly depending on the bean cultivar [45].

Plant-derived essential oils have distinct advantages over synthetic chemicals, e.g., their residues are easily biodegradable in food and the environment [46,47,48,49,50,51]. Besides, thanks to their various mechanisms of action, resistance to these substances does not easily occur [7,52,53]. The mortality of *T. urticae* varied depends on tested concentrations and TSST stages. Tested essential oils (clove, basil, and peppermint oils) were more effective on adults of two-spotted spider mite, *T. urticae* than immature stages because LC50 against adults was lower than immature stages. The current study compared the effects of essential oils and abamectin on egg laying, the life cycle, longevity, and the length of life in the female leaves of the Toshka (SC 349) cultivar cucumber plant. When fed on treated Toshka (SC 349) cultivar discs, the total number of deposited eggs per female was reduced significantly in the treatment of abamectin. Moreover, results showed that abamectin was the most effective compound, followed by clove oil and peppermint oil, whereas basil oil had the least effect on the number of deposited eggs.

Attia et al. [25] evaluated thirty-one essential oils extracted from plants collected from Tunisia as a new approach to controlling *T. urticae,* including *Melia azedarach,* which has the second highest *T. urticae* mortality under laboratory conditions. Different essential oils from 14 species of Lamiaceae caused mortality in immature and adult mites, *T. cinnabarinus* Boisd [54]. Eldoksch, et al. [55] found that the vapors of clove essential oil were the most effective in killing of TSST, while basil and peppermint were the least effective.

Clove oil with its main component eugenol showed comparable effects to abamectin when comparing basil and peppermint EOs. In this regard, Mahmoud and Kassem [56] found that clove essential oil has a considerable effect against *T. urticae*, showing that mortality was high at 24 h and three days post treatment. No effect on mortality percentage was observed after seven days post-treatment. No mortality was noticed in control. The calculated LC50 after seven days of treatment was 0.203% *v*/*v*. The %mortality of adult-females “*T. urticae*” increased with concentration when treated with the clove essential oil. The mechanism of clove EOs involves lowering the Acetylcholine-esterase “AchE” and increasing Alkaline-phosphatase “ALP”, hence killing the pests [57]. Additionally, Archita et al. [58] examined the effectiveness of clove oil against the tea-infesting red spider mite, *Oligonychus coffeae* Nietner (Acari: Tetranychidae) (Acari: Tetranychidae). They discovered that the mortality of *O. coffeae* varied based on the concentrations and durations of the mites’ exposure to the oil following its application. In addition, it has been demonstrated that clove oil at precise quantities is beneficial against adult mites.

Eugenol is the main component of clove essential oil that is effective against mites [59]. This phenylpropanoid has demonstrated potent acaricidal effects on mites, where the double bond’s position in the molecule’s side chain plays a pivotal role in the bioactivity [60,61]. Eugenol is also effective for treating bee products against varroosis [62]. Regarding the mode of action, it has been assumed that its functional group may interfere with the mitochondrial respiration of the target mite [63,64].

On the other hands, Abd-Allah et al. [65] showed that mint derivatives, especially menthol and mint extract, were effective against *T. cinnabarinus* and *T. urticae* while having little effect on the predator *Neoseiulus* sp. *T. cinnabarinus* had LC50 values of 5713.9, 9631.03, and 13,782.6 ppm for menthol, mint extract, and mint oil, respectively. However, the LC_50_ values for *T. urticae* were 5463.2, 7349.7, and 7334.3 ppm for the previously stated tested substances. The mint derivatives were more effective against *T. urticae* than against *T. cinnabarinus*. Menthol was more effective than the other mint derivatives against the predator *Neoseiulus* sp. No studies discussed the anticardia effect of basil oil, and based on our results it has the lowest effects against *T. urticae.*

The mode of action of plant essential oils and their bioactive components was suggested by Enan [66], who indicated that the toxicity of constituents of essential oils against insect pests might be related to the octopaminergic nervous system of insects. Another suggestion is that some monoterpenes may inhibit cytochrome P450-dependent monooxygenases [67]. Ryan and Byrne [68] found a link between the toxicity of monoterpenes, their ability to inhibit acetylcholinesterase (AChE), and their ability to kill insects or ticks.

Generally, from the obtained results, it can be concluded that the immature development time, longevity, life span, and life cycle of *T. urticae* varied when reared on Toshka (SC 349) treated with essential oils and abamectin in vitro conditions

## 5. Conclusions

The TSSM has adverse effects on crops. In this study, Toshka (SC 349) was selected as the most suitable cultivar to increase the mite reproduction, regarding immature time and other biological aspects. Furthermore, it is necessary to test the host suitability of many cucumber cultivars in the hope of discovering resistant cultivars that can be used safely in the context of integrated pest management against *T. urticae*. It is worthwhile to mention that clove oil and its main component eugenol have a comparable impact to abamectin, followed by peppermint oil, whereas basil oil had the least effect on the number of deposited eggs, in addition to eco-friendly pesticides extracted from plants or other bioactive natural compounds with low toxicity, as well as recent technology or proper tactics to achieve sustainable mite control either in vitro or under field conditions.

## Figures and Tables

**Table 1 life-12-01751-t001:** Oil yield of tested EOs.

EOs Name	Family Name	Used Part	Oil Yield (%)
Clove (*Syzygium aromaticum*)	Myrtaceae	buds	2
Peppermint, (*Mentha piperita*)	Lamiaceae	Leaves	1.5
Basil, (*Ocimum basilicum*)	Lamiaceae	Leaves	0.8

**Table 2 life-12-01751-t002:** Chemical composition of purple basil essential oil.

	Components	RT (min)	% Area in GC-FID	KI
1	a-Pinene	11,883	0.41	58.3
2	Camphene	13,750	0.01	45
3	b-Pinene	15,817	1.45	81.7
4	β-Myrcene	19,318	0.46	431.8
5	Bornylene	20,283	0.28	228.3
6	Eugenol	20.62	3.45	262
7	1.8-Cineole	21,133	6.18	313.3
8	2-Hexenal	21,160	0.04	216
9	Sulcatone	26,850	0.01	585
10	Fenchone	29,750	0.01	475
11	3-Methyl-hepta-1,6-dien-3-ol	30,233	0.01	523.3
12	b-Thujone	30,858	3.27	285.8
13	Caprylyl-acetate	33,367	0.05	536.7
14	2,4-Heptadienal	33,800	0.04	580
15	Linalool	36,750	32.5	775
16	Germacrene D	37,867	0.14	886.7
17	a-Bergamotene	38,350	3.16	835
18	Terpinene-4-ol	38,497	0.03	849.7
19	Benzeneacetaldehyde	39,900	0.04	490
20	Borneol L	42,435	0.38	743.5
21	delta-Guaiene	43,133	0.51	813.3
22	Geraniol	47,667	0.04	1066.7
23	Methylcinnamate	56,533	41.39	1453.3
24	a-Cadinol	58,833	4.5	1383.3
	Total identified (%)		98.36	

**Table 3 life-12-01751-t003:** Chemical compositions of EOs from the clove buds.

	Components	RT (min)	% Area in GC-FID	KI
1	Eugenol	20.64	50.2	264
2	Isoeugenol	20.7	16.7	270
3	Caryophyllene	22.22	19.3	322
4	Humulene	23.12	3.5	412
5	α-Amorphene	23.78	0.5	478
6	Acetyleugenol	24.96	7.6	396
	Total identified (%)		97.8	

**Table 4 life-12-01751-t004:** Chemical compositions of EOs from the peppermint plants.

	Components	RT (min)	% Area in GC-FID	KI
1	α-pinene	11.88	0.39	58
2	Sabinene	13.79	0.2	279
3	α-Terpinene	16.13	0.11	413
4	γ-Terpinene	18.33	0.3	633
5	1,8-Cineole	21.13	4.1	613
6	Menthone	23.60	22.82	860
7	Mentho furan	23.90	11.79	890
8	Menthol	24.81	42.13	981
9	Iso-Menthol	25.23	1.89	623
10	NeoisoMenthol	25.37	0.99	637
11	Pulegone	27.65	2.39	665
12	Neo-Menthyl acetate	28.92	1.3	792
13	Piperitone	28.46	0.5	746
14	Menthyl acetate	29.77	4.35	877
15	E-Caryophyllene	35.51	1.85	751
16	Germacrene D	38.16	2.9	1016
	Total identified (%)		98.01	

**Table 5 life-12-01751-t005:** Mean developmental periods (in days) of *Tetranychus urticae* female and sex ratio when reared on leaves of three cucumber cultivars at 27 ± 3 °C and 70 ± 5% R.H.

Cucumber Cultivar	Immature Development Time (Days)					Longevity	Generation Time	Life Span	Sex Ratio
	Egg	Larva	Protonymoh	Deutonymph	Total Immature Time	(Days)	(Days)	(Days)	(Male: Female)
Chief (SC 4145)	3.34 ± 0.05 a	3.41 ± 0.07 b	3.25 ± 0.04 b	2.94 ± 0.09 ab	12.99 ± 0.25 a	14.98 ± 1.55 a	27.97 ± 1.72 a	14.90 ± 0.45 a	1:4.12 *
Raian (CB898)	3.15 ± 0.04 b	3.23 ± 0.06 b	3.21 ± 0.04 b	2.92 ± 0.09 ab	12.30 ± 0.23 b	13.95 ± 1.64 b	26.25 ± 1.97 ab	13.85 ± 0.41 b	1:4.37 *
Toshka (SC 349)	3.10 ± 0.03 b	3.16 ± 0.5 a	3.06 ± 0.06 a	2.71 ± 0.08 b	12.24 ± 0.22 b	12.88 ± 2.04 b	25.12 ± 2.68 ab	13.59 ± 0.40 b	1:5.71 *

Note: Each value is a mean of 10 replicates; Means followed by similar letters in column are not significantly different (one-way ANOVA, *p* ≤ 0.05). * Significant difference with expected ratio of 1:1 (*χ*^2^, *p* < 0.05).

**Table 6 life-12-01751-t006:** Longevity (in days) of *Tetranychus urticae* when reared on leaves of three cucumber cultivars at 27 ± 3 °C and 70 ± 5% R.H.

Cultivars	Pre-Oviposition Period	Oviposition Period	Post-Oviposition Period
Chief (SC 4145)	1.91 ± 0.09 a	10.63 ± 0.21 a	2.44 ± 0.08 a
Raian (CB898)	1.55 ± 0.07 b	9.96 ± 0.18 b	2.39 ± 0.09 b
Toshka (SC 349)	1.35 ± 0.05 c	9.21 ± 0.13 c	2.27 ± 0.06 c
L.S.D 0.05	0.243	0.628	0.345

Note: ±(SE): Standard error.; Each value is a mean of 10 replicates. Lowercase letters in each column indicate the significant differences (*p* ≤ 0.05).

**Table 7 life-12-01751-t007:** Efficacy of different concentrations of essential oils on adults of two—spotted spider mite *T. urticae* reared on cucumber plant Toshka (SC 349) after 24, 48 and 72 h.

Concentration (µLL^−1^ Air)	% Mortality, LC50, LC90, Confidence Limits and Slop Values 24 and 48 h Post-Treatment								
	Clove Oil			Basil Oil			Peppermint Oil		
	24 h	48 h	72 h	24 h	48 h	72 h	24 h	48 h	72 h
0.0 (control)	0.00 ± 0.0 e	2.00 ± 0.1 e	4.00 ± 0.2 d	0.00 ± 0.0 e	2.00 ± 0.2 e	4.00 ± 0.1 e	0.00 ± 0.0 e	2.00 ± 0.0 e	4.00 ± 0.6 d
50.0	55.80 ± 0.1 d	58.60 ± 0.2 d	62.20 ± 0.1 c	24.60 ± 0.1 d	32.20 ± 0.7 d	38.20 ± 0.0 d	34.20 ± 0.6 c	44.60 ± 0.3 d	48.80 ± 0.8 c
100.0	68.60 ± 0.3 c	72.00 ± 0.5 c	76.00 ± 0.3 b	36.40 ± 0.2 c	46.80 ± 0.1 c	54.60 ± 0.2 c	46.60 ± 0.7 b	56.80 ± 0.6 c,d	62.40 ± 0.5 bc
150.0	78.80 ± 0.6 b	84.20 ± 0.9 b	90.00 ± 0.6 a,b	50.20 ± 0.8 b	62.60 ± 0.2 b	66.80 ± 0.4 b	58.80 ± 0.9 b	64.20 ± 0.0 c	68.60 ± 0.2 b,c
200.0	100.00 ± 0.0 a	100.00 ± 0.0 a	100.00 ± 0.6 a	62.40 ± 0.5 a	70.80 ± 0.8 a,b	76.00 ± 0.9 a,b	78.20 ± 0.2 a,b	80.20 ± 0.9 b	84.80 ± 0.6 b
400.0	100.00 ± 0.0 a	100.00 ± 0.0 a	100.00 ± 0.0 a	68.80 ± 0.2 a	80.40 ± 0.7 a	86.20 ± 0.1 a	88.60 ± 0.8 a	92.40 ± 0.1 a	96.60 ± 0.8 a
LC50	46.80			138.40			68.80		
(Confidence limits)	(36.60–42.40)			(106.80–1650.20)			(102.80–120.20)		
LC90	150.00			442.20			325.00		
(Confidence limits)	(197.20–228.80)			(396.20–456.40)			(246.40–388.60)		
Slope value ± SE	5.21 ± 0.72			1.59 ± 0.23			1.66 ± 0.66		

Mean values followed by different letters in the same column differ significantly at *p* ≤ 0.05 level according to Duncan’s multiple range test.

**Table 8 life-12-01751-t008:** Efficacy of different concentrations of essential oils on immature stages of two—spotted spider mite *T. urticae* reared on cucumber plant Toshka (SC 349) after 24, 48 and 72 h.

Concentration (µLL^−1^ Air)	Mean % Mortality, LC50, LC90, Confidence Limits and Slop Values 24 and 48 h Post-Treatment								
	Clove Oil			Basil Oil			Peppermint Oil		
	24 h	48 h	72 h	24 h	48 h	72 h	24 h	48 h	72 h
0.0 (control)	0.00 ± 0.0 d	2.00 ± 0.1 e	4.00 ± 0.1 e	0.00 ± 0.0 e	2.00 ± 0.3 f	4.00 ± 0.1 e	0.00 ± 0.2 f	2.00 ± 0.3 f	4.00 ± 0.2 e
50.0	42.40 ± 0.1 b,c	54.40 ± 0.6 d	60.20 ± 0.4 d	22.20 ± 0.1 d	30.20 ± 0.1 e	36.40 ± 0.2 d	30.60 ± 0.0 e	42.40 ± 0.2 e	48.00 ± 0.3 d
100.0	56.80 ± 0.2 b	66.00 ± 0.8 c	72.00 ± 0.1 c	30.60 ± 0.2 c	42.40 ± 0.3 d	50.80 ± 0.5 c	42.60 ± 0.2 d	54.40 ± 0.9 d	60.20 ± 0.5 c,d
150.0	74.80 ± 0.6 a,b	80.20 ± 0.3 b	88.00 ± 0.2 ab	48.20 ± 0.5 b	60.20 ± 0.8 b,c	66.00 ± 0.9 b,c	56.80 ± 0.1 c	62.60 ± 0.2 c	66.60 ± 0.3 c
200.0	80.20 ± 0.1 a	92.20 ± 0.3 a	100.00 ± 0.0 a	56.40 ± 0.4 a,b	68.80 ± 0.9 b	74.40 ± 0.2 b	70.20 ± 0.3 b	74.20 ± 0.3 b	84.40 ± 0.8 b
400.0	96.60 ± 0.7 a	100.00 ± 0.5 a	100.00 ± 0.0 a	60.20 ± 0.8 a	78.00 ± 0.2 a	84.60 ± 0.1 a	82.20 ± 0.8 a	86.40 ± 0.4 a	94.40 ± 0.3 a
LC50	58.80			158.80			125.40		
(Confidence limits)	(54.60–68.20)			(156.00–186.40)			(142.20–168.60)		
LC90	186.72			396.20			296.20		
(Confidence limits)	(197.20–228.80)			(426.20–456.40)			(440.40–520.60)		
Slope value ±SE	5.75 ± 0.8			1.42 ± 0.2			1.88 ± 0.4		

Mean values followed by different letters in the same column differ significantly at *p* ≤ 0.05 level according to Duncan’s multiple range test.

**Table 9 life-12-01751-t009:** Effect of different compound’s residues on egg of. *T. urticae* on leaf discs of cucumber plant Toshka (SC 349) cultivar.

Tested Materials	No. of Eggs Deposited/5 Adult Females						LSD0.01
	1st Day	2nd Day	3rd Day	4th Day	5th Day	Generalmean	
Control (Toshka (SC(349)	24.20 ± 0.82 a,E	25.40 ± 0.95 a,D	25.80 ± 0.50 a,D	28.60 0.26 a,B	34.20 ± 0.90 a,A	27.74 ± 0.68 a,C	1.36
Clove oil	13.00 ± 0.82 c,E	15.25 ± 0.86 c,D	18.60 ± 0.59 c,C	20.22 ± 0.22 e,B	26.80 ± 0.74 d,A	18.76 ± 0.64 d,C	1.22
Basil oil	16.80 ± 0.44 b,E	19.20 ± 0.76 b,D	21.60 ± 0.60 b,C	24.80 ± 0.36 b,B	32.00 ± 0.55 b,A	22.88 ± 0.54 b,,C	1.09
Peppermint oil	13.00 ± 0.62 c,E	15.60 ± 0.66 c,D	18.80 ± 0.75 c,C	23.20 ± 0.44 c,B	29.20 ± 0.52 c,A	19.96 ± 0.59 c,C	1.16
Abamectin	11.00 ± 0.52 d,E	12.20 ± 0.66 d,D	17.60 ± 0.58 d,C	21.20 ± 0.32 d,B	27.00 ± 0.72 d,A	17.80 ± 0.56 d,C	1.10

Data are presented mean ± SD; Means followed by lowercase letters in same column indicate significant difference between tested materials on eggs. upper letters in the same raw indicate significant difference between the effect of each compound on egg per days at *p* ≤0.05.

**Table 10 life-12-01751-t010:** Comparison between developmental periods (in days) of *Tetranychus urticae* females reared on leaves of cucumber plant Toshka (SC 349) cultivar treated with essential oils and abamectin at 27 ± 3 °C and 70 ± 5% R.H.

Treatments	Immature Development Time (Days)					Adult Longevity	Life Span	Life Cycle
	Egg	Larva	Protonymoh	Deutonymph	Total Immature	(Days)	(Days)	(Days)
Clove oil	3.84 ± 0.06 ab	3.45 ± 0.08 b	3.63 ± 0.04 ab	3.14 ± 0.05 ab	14.06 ± 0.23 b	16.82 ± 1.55 a	30.88 ± 1.78 b	20.66 ± 1.61 a,b
Basil oil	3.78 ± 0.05 b	3.34 ± 0.07 b	3.46 ± 0.07 b	3.11 ± 0.05 b	13.69± 0.24 c	15.55 ± 1.64 b	29.24 ± 1.88 c	19.33 ± 1.69 b
Peppermint oil	3.78 ± 0.03 b	3.30 ± 0.6 a	3.12 ± 0.04 c	3.02 ± 0.04 b	13.22 ± 0.17 c	15.78 ± 2.04 b	29.00 ± 2.21 c	19.56 ± 2.07 b
Abamectin	4.13 ± 0.07 a	3.58 ± 0.09 a	3.94 ± 0.08 a	3.34 ± 0.09 a	14.99 ± 0.33 a	16.69 ± 2.31 a,b	31.68 ± 2.64 a	20.82 ± 2.38 a

Note: Each value is a mean of 10 replicates; Means followed by similar letters in column are significantly different by LSD tests at *p* ≤ 0.05. Data are average of 10 replicates.

## Data Availability

The data presented in this study are available on request from the corresponding authors.

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
