# Peer review of "Chemical Characterization of Clove, Basil and Peppermint Essential Oils; Evaluating Their Toxicity on the Development Stages of Two-Spotted Spider Mites Grown on Cucumber Leaves"

_life, 2022, doi:10.3390/life12111751_

Round 1
Reviewer 1 Report
Chemical Composition, Toxicity and Biological Properties of Three Essential Oils against Tetranychus urticae Koch (Acari: Tetranychidae) on Cucumber Leaves Cultivars
General Comments: This manuscript addressed to determine effect of basil, peppermint, and clove EOs on the survival, biological aspects, and reproduction of Tetranychus urticae (Acari; Tetranychidae) in vitro. In addition, it evaluates the development of the pest in three Cucumber cultivars and uses the cultivar with the best mite fitness to test the effects of EOs. It is an important topic and one of general interest; however, it has many aspects to improve. Although the work shows numerous and interesting experiments, they are not reflected in the objectives or in the discussion. The discussion must necessarily be rewritten since it is very concise and leaves out the discussion about most of the experiences, focusing on the differences between cultivars making very little reference to what I consider to be the most novel research in the work (which is even mentioned in the title and objective) and is the toxicity and biological properties of the three essential oils tested. There is a lack of reorganization in each part of the work, for example the objectives that are mentioned in the introduction, are lost later, when they are poorly developed in the discussion. The soul of the work is not the difference between cucumber cultivars, but mainly the use of EOs for mite control and this should be very clear. Even the title should be reformulated, since EOs were never tested on all cultivars, but only in one. The selected cultivar in which the oils are finally tested is the one with the best fitness for the mite, but that only serves for the selection and nothing else, it should not be the center of the work. I understand that abamectin is a compound of known activity currently used to control the pest; therefore it is not a result to say that it is good, since it is already a positive control. It is interesting to compare the mortality produced by the oils with this positive control, but only to evaluate how active the chosen oils are. The manuscript has reviewed by a native English speaker.
Some specific comments are reported below, referring to line numbering of the manuscript.
Specific Comments:
Line 2-4 change the title, just as it is written, refers to the fact that the oils are tested in various cultivars, something that is not true. The title should reflect the main objective of the work where oils are tested at different development stages of mite.
Line 23-38. Re-write the abstract, highlighting the objectives of the work and its main results.
Line 27- put scientific name in italics
Line 32- LC50 and LC90
Line 34- According to the LC50 obtained, adults are more susceptible than immature states. Check these results.
Introduction
The introduction lacks a brief mention of the mite cycle, which is very important if the different stages of the mite are tested. Likewise, it would be important to include in the introduction what abamectin is and why it is used to compare with the EOs. Why EOs can be used for control mite instead of abamectin? Does this compound generate resistance? What is the disadvantage of its use?
Line 45 Natural products such as essential oils (EOs)….. and line 54- …Two-spotted Spider Mites (TSSM)….. Once both abbreviations are defined, it is advisable to use them throughout the text.
Line 69- Tetranychus urticae
Materials and methods
Given the large number of experiences that are carried out, a diagram is suggested as an illustration of the different activities that are carried out. In addition, the materials and methods should be rewritten to better understand the experiences carried out. For example, the calculation of LC50 must be mentioned within the toxicity tests, since it was calculated for both adults and immature stages. Why Toshka cultivar was selected to analyze the effects of EOs on the mites?
Line 86- put scientific name in italics
Table 1. put scientific name in italics
Line 149- clove bud EOs…
Line 165- what standard solution was used?
Line 171-174- mortality was calculated not only at 24 and 48 h but also at 72 h. Please rewrite this paragraph to make it consistent.
Line 177- what concentration of abamectin was used?
Line 182- The leaves…
Line 186- How many ml of solution was applied?
Line 197- How was the LC50 calculated? Was Probit regression used?. LSD test was used in the table 10 and chi squared in the table 5. Please detail these analyses.
Results
Check the values of the tables, many do not match what is mentioned in the text. In addition, the letters of significance of the tables, in some cases are lost.
Line 201- change “chemical analyses” for “oil yield”
Line 203-211- check the results. The data in the table does not match the data in the text.
Table 2, 3 and 4 could be combined in a single table
The longevity data of table 5 are repeated in table 6. Table 5 and 6 should be combined in a single table
Table 6. Check the letters of significance
Line 296- The effect of clove, basil and peppermint EOs
Table 7 and table 8. Standard errors of mortality percentages are missing. Why the LC50 and LC90 were calculated at 48h? This data is usually calculated for 24 hours.
Line 319- put scientific name in italics
Line 346- …The effects of EOs (concentration 400 units?), stock solution (SS). What was the stock solution?
Table 9. Check the letters of significance. If you want to make comparisons between columns or between rows within the table, you must choose two different fonts for such comparisons. For example, uppercase letters for comparisons between rows and lowercase letters for comparisons between columns.
Line 379-382- this is part of the discussion not results
Table 10. Does it have any basis to compare between columns? I would only compare within the column, to see differences in effect between the compounds tested.
Discussion
The discussion needs to be rewritten. The discussion is very brief and poor. Many very relevant results of the work are not commented on in this section. Minor issues are discussed but the most important of the objectives, for example, the analysis of mortality and the effects that the different EOs tested produce in the different stages of the mites, are not addressed or are briefly addressed. Data from the results are repeated in the discussion for example in line 398.
Line 391-396 improve paragraph writing
Line 420-421. According to the LC50 shown in Table 7 and 8, adults would be more susceptible to the EOs tested than immature states. Check this data.
Line 433-434. Dangerous in what sense? mortality? development? Reproduction of TSST? Detail.
Line 435- Why is it chosen to discuss on eugenol and not the other main compounds of the EOs tested?
Line 441-446. Very general paragraph regarding the mode of action of the main compounds of the EOs. I would have expected something more specific from eugenol that seems to be the most active compound of clove.
Rewrite the conclusion. The main results and objectives should be reflected. Which essential oil is the best one to combat the pest and why would you use it?
Author Response
Reviewer 1
Comments and Suggestions for Authors
Chemical Composition, Toxicity and Biological Properties of Three Essential Oils against Tetranychus urticae Koch (Acari: Tetranychidae) on Cucumber Leaves Cultivars
General Comments: This manuscript addressed to determine effect of basil, peppermint, and clove EOs on the survival, biological aspects, and reproduction of Tetranychus urticae (Acari; Tetranychidae) in vitro. In addition, it evaluates the development of the pest in three Cucumber cultivars and uses the cultivar with the best mite fitness to test the effects of EOs. It is an important topic and one of general interest; however, it has many aspects to improve. Although the work shows numerous and interesting experiments, they are not reflected in the objectives or in the discussion. The discussion must necessarily be rewritten since it is very concise and leaves out the discussion about most of the experiences, focusing on the differences between cultivars making very little reference to what I consider to be the most novel research in the work (which is even mentioned in the title and objective) and is the toxicity and biological properties of the three essential oils tested. There is a lack of reorganization in each part of the work, for example the objectives that are mentioned in the introduction, are lost later, when they are poorly developed in the discussion. The soul of the work is not the difference between cucumber cultivars, but mainly the use of EOs for mite control and this should be very clear. Even the title should be reformulated, since EOs were never tested on all cultivars, but only in one. The selected cultivar in which the oils are finally tested is the one with the best fitness for the mite, but that only serves for the selection and nothing else, it should not be the center of the work. I understand that abamectin is a compound of known activity currently used to control the pest; therefore it is not a result to say that it is good, since it is already a positive control. It is interesting to compare the mortality produced by the oils with this positive control, but only to evaluate how active the chosen oils are. The manuscript has reviewed by a native English speaker.
Response: We thank the reviewer for the effort in reviewing and enhancing the manuscript. The title, introduction and discussion were reformulated to meet the study objectives.
Specific Comments:
Line 2-4 change the title, just as it is written, refers to the fact that the oils are tested in various cultivars, something that is not true. The title should reflect the main objective of the work where oils are tested at different development stages of mite.
Response: Thanks for the reviewer. The title was reformulated to meet the study objectives
Line 23-38. Re-write the abstract, highlighting the objectives of the work and its main results.
Response: Thanks for the reviewer, the abstract was rewritten accordingly
Line 27- put scientific name in italics
Response: Done as requested
Line 32- LC50 and LC90
Response: Done as requested
Line 34- According to the LC50 obtained, adults are more susceptible than immature states. Check these results.
Response: Thanks for the reviewer, it was adjusted accordingly
Introduction
The introduction lacks a brief mention of the mite cycle, which is very important if the different stages of the mite are tested. Likewise, it would be important to include in the introduction what abamectin is and why it is used to compare with the EOs. Why EOs can be used for control mite instead of abamectin? Does this compound generate resistance? What is the disadvantage of its use?
Response: Thanks for the reviewer for enhancing the manuscript, the introduction reformulated and these points were considered
Line 45 Natural products such as essential oils (EOs)….. and line 54- …Two-spotted Spider Mites (TSSM)….. Once both abbreviations are defined, it is advisable to use them throughout the text.
Response: Done as requested
Line 69- Tetranychus urticae
Response: Done as requested
Materials and methods
Given the large number of experiences that are carried out, a diagram is suggested as an illustration of the different activities that are carried out. In addition, the materials and methods should be rewritten to better understand the experiences carried out. For example, the calculation of LC50 must be mentioned within the toxicity tests, since it was calculated for both adults and immature stages. Why Toshka cultivar was selected to analyze the effects of EOs on the mites?
Response: Toshka planted in the fields of Belles Province, Sharkia Governorate and highly infected
Line 86- put scientific name in italics
Response: Done as requested
Table 1. put scientific name in italics
Response: Done as requested
Line 149- clove bud EOs…
Response: Done as requested
Line 165- what standard solution was used?
Response: Standard solution of all essential oils
Line 171-174- mortality was calculated not only at 24 and 48 h but also at 72 h. Please rewrite this paragraph to make it consistent.
Response: Done as requested
Line 177- what concentration of abamectin was used?
Response: Recommended rate (RC) of 15 mL/100L
Line 182- The leaves…
Response: Done as requested
Line 186- How many ml of solution was applied?
Response: We use one ml
Line 197- How was the LC50 calculated? Was Probit regression used?. LSD test was used in the table 10 and chi squared in the table 5. Please detail these analyses.
Response: The statistical analysis was reformulated
Results
Check the values of the tables, many do not match what is mentioned in the text. In addition, the letters of significance of the tables, in some cases are lost.
Response: The values were checked and corrected as Tables
Line 201- change “chemical analyses” for “oil yield”
Response: Done as requested
Line 203-211- check the results. The data in the table does not match the data in the text.
Response: The values were checked and corrected as Tables
Table 2, 3 and 4 could be combined in a single table
Response: Thanks for the reviewer for this suggestion, but combining them will be confused
The longevity data of table 5 are repeated in table 6. Table 5 and 6 should be combined in a single table
Response: Thanks for the reviewer for this suggestion, we deleted longevity from Table 6 and keep it separated
Table 6. Check the letters of significance
Response: Thanks for the reviewer, lowercase letters were added to indicate the statistical differences
Line 296- The effect of clove, basil and peppermint Eos
Response: Done as requested
Table 7 and table 8. Standard errors of mortality percentages are missing. Why the LC50 and LC90 were calculated at 48h? This data is usually calculated for 24 hours.
Response: We added standard error in these Tables. We calculated three times at interval of 24, 48, and 72 as experimental design and indicate the efficiency of using essential oils for several days of treatment
Line 319- put scientific name in italics
Response: done
Line 346- …The effects of EOs (concentration 400 units?), stock solution (SS). What was the stock solution?
Response: It was stock solutions of EOs as mentioned in materials and methods sec 2.4
Table 9. Check the letters of significance. If you want to make comparisons between columns or between rows within the table, you must choose two different fonts for such comparisons. For example, uppercase letters for comparisons between rows and lowercase letters for comparisons between columns.
Response: The statistical letters were checked and corrected
Line 379-382- this is part of the discussion not results
Response: It was moved to discussion
Table 10. Does it have any basis to compare between columns? I would only compare within the column, to see differences in effect between the compounds tested.
Response: The statistical letters within column were checked and corrected
Discussion
The discussion needs to be rewritten. The discussion is very brief and poor. Many very relevant results of the work are not commented on in this section. Minor issues are discussed but the most important of the objectives, for example, the analysis of mortality and the effects that the different EOs tested produce in the different stages of the mites, are not addressed or are briefly addressed. Data from the results are repeated in the discussion for example in line 398.
Response: The discussion was reformulated and more explanations for results were added
Line 391-396 improve paragraph writing
Response: Done as requested
Line 420-421. According to the LC50 shown in Table 7 and 8, adults would be more susceptible to the EOs tested than immature states. Check this data.
Response: Immature stages of two- spotted spider mite T. Urticae than were more sensitive than adults
Line 433-434. Dangerous in what sense? mortality? development? Reproduction of TSST? Detail.
Response: Mortality
Line 435- Why is it chosen to discuss on eugenol and not the other main compounds of the EOs tested?
Response: Eugenol is the main compound in clove oil content and the clove oil has a good effect
Line 441-446. Very general paragraph regarding the mode of action of the main compounds of the EOs. I would have expected something more specific from eugenol that seems to be the most active compound of clove.
Response: We add the specific mechanism of eugenol in discussion
Rewrite the conclusion. The main results and objectives should be reflected. Which essential oil is the best one to combat the pest and why would you use it?
Response: The conclusion was reformulated accordingly
Reviewer 2 Report
The paper presents the insecticidal activity of three essential oils (clove, peppermint and basil) on Tetranychus urticae Koch. The results regarding the quantitative chemical composition of essential oils can be improved by GC-MS and GC-FID analysis and the calculation of Kovats index (KI). The distillation time (8 h) for peppermint and basil is too long!!! Also, the 1,8-cineole, menthol and eugenol are not phenols (L49).
Author Response
Reviewer 2
Comments and Suggestions for Authors
The paper presents the insecticidal activity of three essential oils (clove, peppermint and basil) on Tetranychus urticae Koch. The results regarding the quantitative chemical composition of essential oils can be improved by GC-MS and GC-FID analysis and the calculation of Kovats index (KI). The distillation time (8 h) for peppermint and basil is too long!!! Also, the 1,8-cineole, menthol and eugenol are not phenols (L49).
Response: Thanks for the reviewer for enhancing the manuscript; % area in GC-FID was clarified, and the retention index was added to all GC-MS Tables
Round 2
Reviewer 1 Report
The revised manuscript is improved and more clearly presents the information. However I find several problems and inconsistencies, mainly in the interpretation of some results, due to an error in the definition of LC50. Please check the statistics used to analyze the data. These errors in the results directly affect the conclusions and on the overall quality of the research.
I highlighted in the manuscript some comments.

Author Response
Response letter
The revised manuscript is improved and more clearly presents the information. However, I find several problems and inconsistencies, mainly in the interpretation of some results, due to an error in the definition of LC50. Please check the statistics used to analyze the data. These errors in the results directly affect the conclusions and on the overall quality of the research.
Response: Thanks for the reviewer for his efforts in enhancing the manuscript. All comments were considered accordingly.
I highlighted in the manuscript some comments.
Response: Thanks for the reviewer. All comments in PDF were responded accordingly as follows
Page 4 comments, Italic removed from Basil; “s” deleted from “buds”
Page 5 comments the definition of LC50 was adjusted, and “lowest” was deleted.
Page 8 comment, the presentation of results in Tables 7 and 8 were enhanced.
Page 10 comments, the statistical analysis of Table 9 was checked and adjusted.
Page 11 comments were adjusted accordingly.